# Multi-Activity Step Counting Algorithm Using Deep Learning Foot Flat Detection with an IMU Inside the Sole of a Shoe

**DOI:** 10.3390/s24216927

**Published:** 2024-10-29

**Authors:** Quentin Lucot, Erwan Beurienne, Michel Behr

**Affiliations:** 1LBA UMR T24, Université Gustave Eiffel, Aix-Marseille Université, 13015 Marseille, France; 2Wizwedge SARL, 13004 Marseille, France; 3Context’, Tyyny, 42500 Le Chambon-Feugerolles, France

**Keywords:** step counting, IMU, gait analysis, deep learning, pedometer, LSTM

## Abstract

Step counting devices were previously shown to be efficient in a variety of applications such as athletic training or patient’s care programs. Various sensor placements and algorithms were previously experimented, with a best mean absolute percentage error (MAPE) close to 1% in simple mono-activity walking conditions. In this study, an existing running shoe was first instrumented with an inertial measurement unit (IMU) and used in the context of multi-activity trials, at various speeds, and including several transition phases. A total of 21 participants with diverse profiles (gender, age, BMI, activity style) completed the trial. The data recorded was used to develop a step counting algorithm based on a deep learning approach, and further validated against a k-fold cross validation process. The results revealed that the step counts were highly correlated to gyroscopes and accelerometers norms, and secondarily to vertical acceleration. Reducing input data to only those three vectors showed a very small decrease in the prediction performance. After the fine-tuning of the algorithm, a MAPE of 0.75% was obtained. Our results show that such very high performances can be expected even in multi-activity conditions and with low computational resource needs making this approach suitable for embedded devices.

## 1. Introduction

Daily step counting, i.e., using devices such as pedometers, was shown to be efficient for public health activity management programs [1,2] as step count monitoring increased physical activity levels [3] and improved various health outcomes such as all-cause mortality or cardiovascular disease [4,5,6]. Even minor increases in step counts such as an additional 1000 steps per day have been linked to important risk reductions in all-cause mortality (6–36%) and cardiovascular disease (5–21%) [4]. For specific populations like the elderly, physical activity has been linked to higher levels of functional health, lower risks of falling, and improved cognitive health, emphasizing the benefits of achieving higher daily step counts [7].

Accurate step detection and counting is also very useful in another field of application, indoor pedestrians tracking; it is a fundamental input of pedestrian dead reckoning (PDR) algorithms, assisting in estimating stride length and walking direction [8,9,10,11].

Traditional step counting methods commonly use inertial measurement units (IMU) such as accelerometers, gyroscopes, and magnetometers [12,13,14,15].

Step counting devices, whether integrated into smartphones, watches, or standalone units, vary in accuracy and efficiency [16]. Studies have shown that wearable devices can accurately track step counts under well-controlled conditions [17,18]. However, accuracy may be altered by the type of activity being performed [19,20] and by environmental conditions [18]. Personalized approaches can still enhance accuracy across different individuals [21], as further discussed.

Various methodologies are available and detailed in the literature. One method is known as thresholding, where a specific acceleration threshold is set for step detection based on specific a priori motion patterns. As an example, several commercial pedometers track steps through hip motion [22]. The main challenge with thresholding algorithms for step counting and detection lies in accurately discriminating steps from other movements, particularly in scenarios with a wide variety of walking patterns, or in scenarios where the subject performs various activities other than walking. Using dynamic thresholding and feature adaptation may help to overcome this challenge [23]. Combining multiple thresholding algorithms may also improve the detection of steps amongst other activities [24]. However, all of these thresholding algorithms need an adjustment of their amplitude thresholds based on the user characteristics, which can impede practical usability [25].

Another step counting method involves frequency analysis, determining step counts by analyzing accelerometer data patterns [26,27].

Rule-based methods are also used, setting predefined rules to identify steps based on specific criteria, which can be customized for different user characteristics or scenarios to enhance step counting accuracy [28]. Rule-based algorithms often involve complex signal processing with numerous thresholds and rules, leading to challenges in practical implementation [29]. The complexity of these algorithms can hinder their usability and may require manual adjustments based on specific user characteristics, impacting their reliability [24]. It can also introduce inaccuracies, especially at low walking speeds [30].

Recently, deep learning (a subset of machine learning, involves training complex neural networks) was considered for step counting, with personalized models. These methods were reported as highly accurate in step counting when using accelerometer data in mono-activity scenarios [19].

Beyond step detection methods, the location of IMUs on the user body is also critical. The gait information was shown to be more stable and more accurate with foot-mounted sensors rather than trunk or shank-mounted ones [31,32,33]. Considering foot-mounted sensors, the most reliable measurements of gait events were obtained by placing the IMU directly into the midsole under the foot arch, rather than laterally at the ankle, or behind the heel, or on the instep [34].

This study aims at developing and testing an original algorithm that is able to discriminate steps in multi-activity scenarios, using the most promising approach: IMUs embedded into midsole and deep learning algorithms.

## 2. Materials and Methods

In our approach, an existing running shoe was first instrumented and then used in the context of multi-activity trials. The data recorded was used to develop a step counting algorithm. Some of the data were also used for validation purposes.

### 2.1. Instrumented Running Shoe

The shoe models used in this study are Wizwedge Heliums (Wizwedge, Marseille, France). These were chosen because they have a removable space in the sole at the back of the shoe, underneath the heel. This made it easier to integrate the sensors.

Accelerometry and angular speed were recorded with a MetaMotionR IMU (MbientLab, San Jose, CA, USA) with dimensions of 36 mm × 27 mm × 10 mm. This device allowed acquisition at a rate of 200 Hz and offered a measurement range of ±16 g for acceleration and ±2000°/s for gyroscopes. This IMU was chosen because it can be connected in real time to a smartphone via Bluetooth using the proprietary Metabase application, which meets the requirements of the test protocol detailed below.

To integrate the device into the shoe, a PEBA-S component was 3D printed. This component was then placed in the hollowed-out sole at heel level. In this way, the center of the IMU was positioned for a standard size 42 EU shoe, 8 mm below the top layer of the insole in contact with the foot (2 mm of insole thickness with 1 mm of 3D printed PEBA-S, and 5 mm as half of the IMU height) and 9 mm behind the heel center of impact, located at 12% of the sole length, at 32 mm (Figure 1).

### 2.2. Multi-Activity Dataset Acquisition Protocol

#### 2.2.1. Participants

Similar studies have shown populations of varying sizes, ranging from 11 to 30 participants [19,35]. Therefore, the target of this study is set to recruiting at least 20 participants. A total of 21 volunteers were recruited from the staff of the LBA laboratory in Marseille, France. The participants were not representative of any particular category of the population, as they had different profiles in terms of age, height, weight, BMI, and activity levels. Their activity levels were defined by their self-reporting of their weekly frequency of moderate-intensity activities (3.0–5.9 metabolic equivalents) and vigorous-intensity activities (6.0 and higher metabolic equivalents) [36]. Subjects with a history of lower-limb injury in the previous 6 months, who declared trouble walking or running, or who were over 60 years of age were excluded from the study.

#### 2.2.2. Protocol

After signing an informed consent form, the participants were fitted with a pair of instrumented shoes in their own size with an IMU integrated into the right shoe only and carried out an initial acclimation phase consisting of a 3 min free-walking sequence. They were then asked to complete a multi-activity track including walking, running, stomping, high knees, butt kicks, and descending stairs. They all completed the track once. The track is illustrated in Figure 2.

#### 2.2.3. Video Acquisition

A video recording was made to correlate the IMU recordings with the participants’ movements during the trial. To achieve this, a simple Xiaomi Mi 9T (Xiaomi Corporation, Beijing, China) smartphone set to a frequency of 60 fps was used and positioned laterally on the track so that the field of view was wide enough to record the entire trial. The same phone was used to record the IMU data through Bluetooth.

In order to synchronize the time between the video and the signal recorded by the IMU, the participants performed at the very beginning of the trial a total of 3 clear heel strikes against the ground. The third impact corresponded to time 0 in the protocol.

#### 2.2.4. Data Pre-Processing

The video of each participant was processed using Kinovea (Version 0.9.5, 2019), frame by frame, to identify the two transition moments when the foot was stationary relative to the ground, corresponding at the beginning and the end of a phase where the foot was flat. For detection, two markers were manually positioned at the front and rear of the shoe, on the sole. Measured with the software below (or alternately above), a threshold of 3° was considered as a transition. At the end of this sequence, for each participant, a video analysis results table with 3 columns was created: time; the « foot status », which was equal to 0 when the foot was not stationary relative to the ground, 1 when it was stationary relative to the ground, or 2 when the status was indeterminate; and the activity category (0 (standing still), 1 (slow walking), 2 (moderate walking), etc.).

Since the acquisition frequency of the IMU was different from that of the video acquisition (200 Hz vs. 60 fps), the IMU data recording files were simultaneously enriched using a Python (Version 3.9.18, 2023) program to annotate the foot status and activity category at each time step. As a result, this enriched input file contained a total of 11 columns: time, 3-axis accelerations, 3-axis gyroscopes, acceleration norm, gyroscope norm, foot status, and activity type. An example of 1 s of enriched data from raw acquisition values is shown in Figure 3.

Subsequently, the time steps for which the foot status was undetermined were not considered for the deep learning algorithm.

### 2.3. Deep Learning Step Counting Algorithm

#### 2.3.1. Data Reduction

To simplify the algorithm and increase the speed of the learning phase of the model, the volume of the data were reduced as follows. The Pearson correlation between the prediction (foot status) and the input data for the three axes and norms of the gyroscope and accelerometer were estimated individually, and then all the data leading to a correlation below 0.1 were removed. The model’s performances for all the remaining columns were determined, i.e., those that obtained a correlation above 0.1. The performance that would have been achieved with all the input data (including the least correlated ones) to estimate the performance loss due to the reduction in the input data volume was also estimated.

#### 2.3.2. Recurrent Neural Network Based Architecture

Recurrent Neural Networks (RNNs) are a class of neural networks primarily used for processing sequential or temporal data, making them well-suited for our situation. RNNs have loops that allow them to retain a memory of previous information. They are fed with vectorized sequential data. Each neuron in the hidden layer is connected to both the current input and its own previous state to estimate the prediction or final result, in this case, for the foot status based on the processed information.

Here, the RNN architecture used is composed of at least one Long Short-Term Memory (LSTM) many-to-one layer, as detailed in the recent literature [37], followed by a dropout layer to prevent overfitting, then a single-unit dense layer. The whole system was associated with a parameter called « rest delay » (RD) which represented the minimum time after foot flat detection before another foot flat phase can be considered (Figure 4). Any detection during the RD was therefore not considered.

The experiments were implemented with Python using the open-source Keras Github library (Version 3.5.0, 2024).

#### 2.3.3. Performance Criteria

Two performance criteria were considered and derived from a previous study [19]:The mean absolute percentage error at the output of the deep learning algorithm, noted as ${MAPE}_{DL}$, calculated as the absolute difference between the predicted foot status vectors and the observed ones;
$${MAPE}_{DL}=\frac{\left|n_{foot-status,predicted}-n_{foot-status,observed}\right|}{n_{foot-status,observed}}$$The global mean absolute percentage error, noted as ${MAPE}_{G}$, calculated as the absolute difference between the number of steps taken by the volunteer and the number of steps estimated by the algorithm.
$${MAPE}_{G}=\frac{\left|n_{steps,predicted}-n_{steps,observed}\right|}{n_{steps,observed}}$$

To facilitate the reading of performance criteria, they are then expressed as percentages. The estimation of these criteria was based on a k-fold cross-validation approach on the participants, following the procedure detailed hereafter. A total of 7 folds were created, each containing data from 3 participants randomly drawn from 21 people. The cross-validation then proceeded in 7 steps. At each step, a different fold was used as the test and validation set, while the remaining 6 folds were combined to form the training set. For each step, the model was trained on 6 training folds and evaluated on the test fold. This yielded 7 estimates of the model’s performance. The performance of the 7 estimates was then averaged to obtain the overall performance estimate of the model.

#### 2.3.4. Fine-Tuning

An iteration was first launched using the entire input data, with the standard deep learning model fed with initial parameters chosen arbitrarily and listed in Table 1. The parameter input window size is the size of the window of anterior data at the considered moment (history data), used as input data for the recurrent LSTM layers.

As listed in Table 1, there are 9 parameters in total, leading to a design of experiment (DOE) of 3,150,000 possible parameter combinations. RD is not considered in this deep learning phase but will be adjusted at the end of the process as discussed further in this paragraph.

Fine-tuning was then performed using PyHopper (Version 1.0.0, 2022) [38]. PyHopper uses a two-stage Markov Chain Monte Carlo (MCMC) optimization algorithm. First, it randomly explores the search space to identify the best regions of interest. Then, it refines the search locally by adjusting the best parameters, guided by a temperature parameter that gradually reduces randomness and focuses the search. PyHopper allocates 25% of its runtime to random search, with the remaining 75% dedicated to local sampling. The main advantage of PyHopper in our approach is that it simplifies the process by eliminating algorithm selection and offers easy customization. When compared to other optimization algorithms, PyHopper identifies competitive hyperparameter settings and completes the search process the quickest, as it was proven to be over 10 times faster than Optuna and HyperOpt [38].

The objective function of the deep learning model was here set as validation loss reduction. To increase the speed of this process, the optimization was first run on a reduced set of 25% of the randomly selected input data producing a provisional set of optimized parameters. The number of epochs was set to 100, and two early stopping criteria were implemented: (1) a patience criteria set to 10 will stop the learning process if no improvement is observed, (2) the maximal duration of the learning process is set to 5 min.

Optimization was run for a total duration of 12 h on a laptop using 32 GB of RAM and a mobile GPU NVIDIA P620 4 GB (Nvidia Corporation, Santa Clara, CA, USA).

At the end of the process, the provisional set of optimized parameters was used on the full input database (instead of only 25%) leading to an estimate of the ${MAPE}_{DL}$ values. Finally, the rest delay (RD) was adjusted as follows: the ${MAPE}_{G}$ were estimated iteratively with the values of the RD going from 25 ms to 1000 ms by steps of 25 ms in order to keep the best RD values in the end.

## 3. Results

### 3.1. Acquisition Protocol

A total of 13 men and 8 women meeting the inclusion criteria were recruited for the experimental phase. Given the sample size, we decided to adopt a non-gender-specific approach in this study. The characteristics of the individuals are detailed in Table 2. The average values of height, weight, and BMI can be considered typical averages for a normal population, with significant variability in the different characteristics. A total of 10 participants reported no moderate or vigorous physical activity, 8 participants reported an activity frequency of 1 to 2 times per week, 2 participants reported a frequency of 3 to 4 times per week and 1 participant reported an activity frequency of more than 5 times per week.

All 21 sessions were correctly recorded and saved for further analysis. A video analysis allowed the construction of a foot status vector, including a total of 244,215 instances where the foot was not flat, 166,438 instances where the foot was observed to be flat, and 43,943 instances where the status was indicated as undetermined, most often because the foot position was not clearly visible in the video.

The protocol was designed such that participants had to perform 40 walking steps, 40 running steps, and a variable number of steps via other activities or descending stairs (see Figure 2). A total of 4017 steps (of all types) were finally counted across all the videos, averaging 191 steps per participant.

### 3.2. Deep Learning

#### 3.2.1. Reduction in the Data Volume

The Pearson correlation matrix between each of the enriched IMU data vectors is presented in Figure 5. The norms of the gyroscope and acceleration show the highest correlation with the foot status, at 0.64 and 0.41, respectively. The best-correlated acceleration axis is the Z vertical axis (0.16). The best-correlated gyroscope axis is the X-axis, with a low value (0.047).

To reduce the volume of input data, all the vectors with a correlation higher than 0.1 were kept for the learning algorithm, i.e., the two norms and the acceleration along the Z vertical axis.

The performance of the process estimated before fine-tuning, with all the input data and then with only the three highly correlated vectors, is given in Table 3.

#### 3.2.2. Fine-Tuning and Validation

The fine-tuning procedure and k-fold cross-validation led to the set of best parameters listed in Table 4. These parameters were implemented in the RNN structure of the Appendix A.

Varying the RD values revealed a best performance of ${MAPE}_{G}=0.75\left(0.47\right)\%$ for a RD equal to 375 ms, with a detection of 4034 steps out of the 4017 performed by the participants (and identified on the videos). Lower or higher values or RD decreased notably the performance of the process. For example, a RD set to 1000 ms would lead to an increased error of ${MAPE}_{G}=53.02\left(1.83\right)\%$.

## 4. Discussion

In this study, a step counting method based on a deep learning approach was proposed and tested on a panel of 21 participants in a relatively rich and complex experimental context, alternating phases of slow and fast walking, running at different speeds, on-the-spot stepping, physical jerks, and stair descent. This is one of the original components of our protocol, as step counting studies in the literature are mostly based on homogeneous walking phases [39].

In terms of performance, multi-activity conditions involving variable-speed movements and numerous transition phases can be considered unfavorable for a step-counting algorithm, as already reported in the literature [40,41]. Despite this, a ${MAPE}_{G}$ of 0.75% was achieved, leading to an error in counting the steps as low as 0.42% (4034 instead of 4017), with a reduced set of input data limited to the norms of the acceleration and gyroscope, and adding acceleration along the Z vertical axis. Therefore, there was very good accuracy despite the multi-activity and important variations in the movement of the subjects when compared to other studies. It is hazardous to compare these results directly to the literature values, given the complexity and, most of all, the specificity of this experimental protocol. Nonetheless, other step counting attempts could not reach ${MAPE}_{G}$ lower than 1.1% for the most efficient approaches reported in the literature, and for this under-homogenous mono-activity of walking conditions with an IMU placed in the foot/ankle area [42].

Our results also show that the foot status parameter (and thus indirectly of the predictive capability) was mostly correlated to the gyroscope norm, which is consistent with the literature [41]; although, in our case, for unclear reasons, the correlation with the gyroscope x, y, or z axes taken individually was very low.

In addition, our approach showed a relatively small performance loss (0.4%) when only three input parameters, i.e., the norms of acceleration and gyroscope, associated with the acceleration along the Z vertical axis, were considered in the prediction, significantly reducing the prediction computation time and the RNN model’s complexity, and making the approach applicable to lightweight devices in terms of computational capacity. Additionally, our results confirm the interest in merging acceleration and gyroscope data, as suggested in the conclusions of another recent study [41].

Several observations can also be made about the optimized parameters of the RNN model. The low number of layers and nodes per layer is explained by the reduced complexity of the input data, thanks to the efficient procedure of input data volume reduction. An input window size of 125 ms (rather than the maximum value of 250 ms) proved sufficient for the optimal results, also providing significant computational time-saving.

These good performances were accompanied by several limitations. Firstly, the experimental track was flat and horizontal, with no ascending or descending phases (except for the stairs at the beginning of the trial). Before this procedure can be applied in ecological conditions, it will be necessary to evaluate the potentially negative impact of slopes. Since a step is characterized here as a phase where the foot is positioned flat on the ground, the algorithm might be misled when, for example, on a very steep slope, a runner never touches the ground with their heel. However, we hypothesize minimal performance deterioration as long as the slope remains moderate. The absence of turns in our experimental track similarly limits future applications.

Another limitation of this study lies in the specific IMU location, underneath the heel. Other approaches found in the literature highlight the influence of IMU location on performance in mono-activity scenarios with devices easier to integrate on the wrist, waist, or thigh [42]. However, our study shows that placing the sensor underneath the heel achieves excellent performance in multi-activity scenarios. Our approach is therefore only transferable on the condition that the IMU is positioned in the same location.

Finally, one of the interests of deep learning is to be able to enrich the training base as the device is used. Here, the input data requires a very time-consuming analysis of video sequences. This limitation could be advantageously circumvented by associating a markerless motion capture tool. Furthermore, since the algorithm is based on a deep learning approach, it is inherently capable of learning in new conditions, such as new activities, and can be further refined as soon as the model is enriched with data corresponding to these new conditions.

Accurately detecting the steps in ecological multi-activity conditions creates opportunities for numerous future applications, including monitoring the efficiency of training programs or estimating injury risk, or can be extended to the monitoring of different types of gaits in biomimetic robots [43].

## 5. Conclusions

The step counting approach proposed in this study showed very good performance despite varied usage conditions, including multi-activity phases covered at different speeds and numerous transitions. These results are encouraging and pave the way for effective activity discrimination, which will have numerous applications in the fields of sports performance and patient care programs.

## Figures and Tables

**Figure 1 sensors-24-06927-f001:**
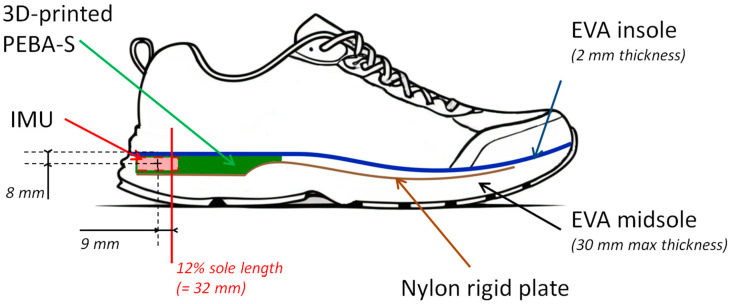
Instrumented wedge and positioning in the shoe sole for a 42 EU size shoe; IMU in red with its location relative to the heel center of impact at 12% sole length, and the top layer of the insole.

**Figure 2 sensors-24-06927-f002:**
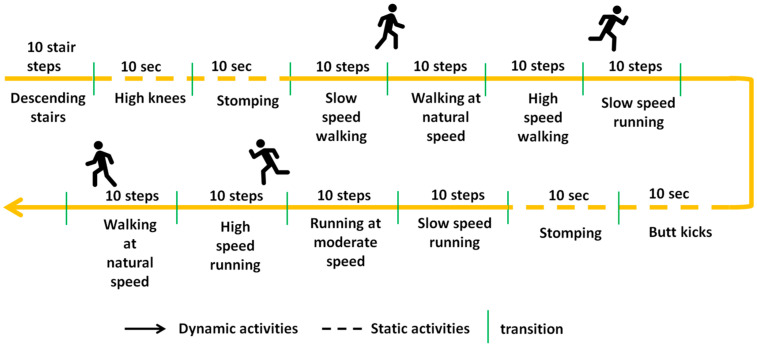
Experimental track (top view).

**Figure 3 sensors-24-06927-f003:**
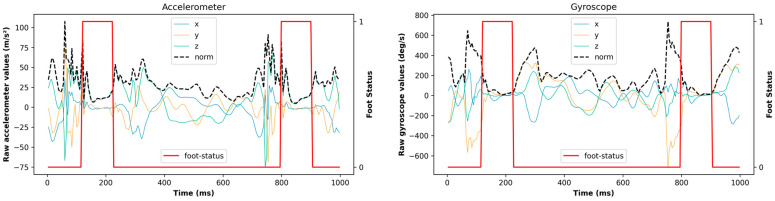
Raw accelerometer (**left**) and gyroscope (**right**) values associated with corresponding foot status.

**Figure 4 sensors-24-06927-f004:**
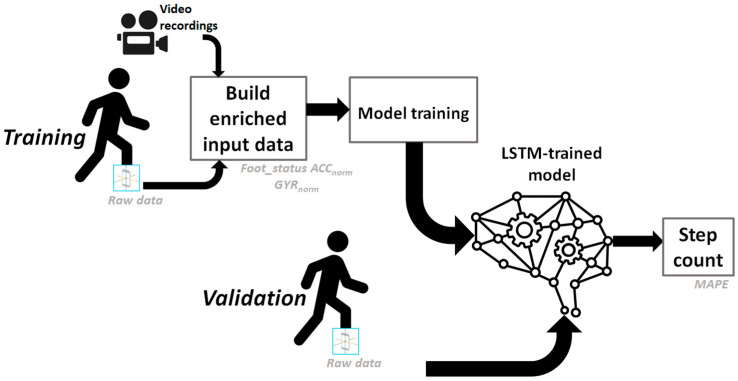
Representation of the complete process of the step detection algorithm using a deep learning approach.

**Figure 5 sensors-24-06927-f005:**
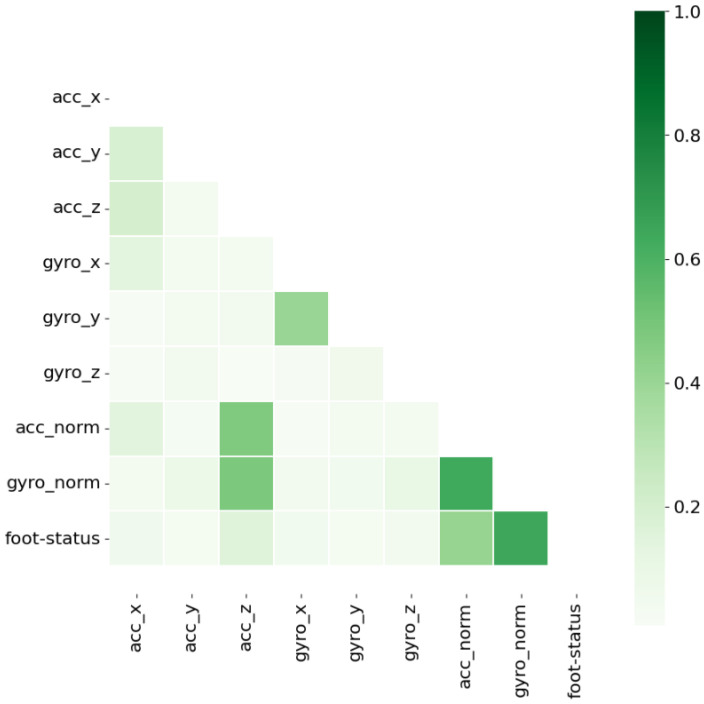
Correlation matrix between the recorded data and the foot status.

**Table 1 sensors-24-06927-t001:** Parameters of the deep learning standard model and the nominal values assigned along with their ranges of variation.

Parameter ID	Range	Number of Values	Nominal Value
Number of LSTM units	[16–128]	5	64
Number of LSTM layers	[1–3]	3	2
Dense layer active	true; false	2	false
Number of dense units	[8–64]	5	N/A
Input window size (ms)	[25–250]	10	50
Dropout	[0.1–0.5]	5	0.4
LSTM activation function	relu; sigmoid; tanh	3	tanh
Learning rate	[10^−2^–10^−4^]	20	10^−3^
Batch size	[64–2048]	7	1024

**Table 2 sensors-24-06927-t002:** Characteristics of the participants.

	Mean	Min	Max	Std
Age (years)	30.5	22	56	9.8
Height (cm)	171	155	186	7.8
Weight (kg)	69.8	43.5	97	13.1
BMI	23.9	18.1	32.3	3.7
Shoe size (EU)	41	37	45.5	2.4

**Table 3 sensors-24-06927-t003:** Performance of the process before fine-tuning.

Performance Criteria	${MAPE}_{DL}$	${MAPE}_{G}$
All data	3.7%	1.7%
AccNorm + GyroNorm + AccZ	4.8%	2.1%

**Table 4 sensors-24-06927-t004:** Optimized parameters of the deep learning algorithm.

Parameter ID	Final Value
Number of LSTM units	32
Number of LSTM layers	2
Dense layer active	False
Number of dense units	N/A
Input window size (ms)	125
Dropout	0.2
LSTM activation function	Tanh
Learning rate	10^−2^
Batch size	1024

## Data Availability

The data presented in this study are available on request from the authors.

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
