# Peer review of "Multi-Activity Step Counting Algorithm Using Deep Learning Foot Flat Detection with an IMU Inside the Sole of a Shoe"

_sensors, 2024, doi:10.3390/s24216927_

Round 1
Reviewer 1 Report
Comments and Suggestions for Authors
In the manuscript "Multi-activity Step Counting Algorithm Using Deep Learning Foot Flat Detection with an IMU Inside the Sole of a Shoe," the authors present several important results. Although the manuscript is well written, the following points must be addressed before it can be accepted for publication:
1. Technical details of the shoe and the position of the sensors within the shoe must be provided to ensure the reproducibility of the experiments. For example, details such as the distance relative to the sole or foot, and the layer thickness between the foot and the sensor should be included.
2. The data provided about the participants should include their gender (male, female) to ensure the reader can assess control over the data used in the study.
3. How representative is the population used in the study? Why was this sample size chosen, and why not more? I believe 21 samples are limited. The selection of these samples must be properly justified.
4. I recommend modifying Figure 1 to provide an enhanced technical diagram, clearly depicting the instrumentation to avoid confusion for the reader.
5. I suggest adding a figure that includes an example of the recorded signal and an explanation of the data's meaning.
6. I recommend including a statement before the conclusion section, highlighting the novelty of the work compared to similar published studies.
7. A paragraph should be added discussing potential future applications of the scientific proposal presented in the manuscript, such as monitoring different types of gaits in quadruped robots for control purposes [Biomimetics 2024, 9, 318].
8. It would be useful for readers if the authors include segments of code (or pseudocode) for the novel algorithms developed in this work.
Author Response
Thank you very much for taking the time to review this manuscript. Please find the detailed responses below and the corresponding revisions in the re-submitted files.
Comment 1: Technical details of the shoe and the position of the sensors within the shoe must be provided to ensure the reproducibility of the experiments. For example, details such as the distance relative to the sole or foot, and the layer thickness between the foot and the sensor should be included.
Response 1: Thanks for pointing this out, the description of the sensor has been improved by adding its dimensions in section 2.1 with the following sentence: ‘Accelerometry and angular speed were recorded with a MetaMotionR IMU (MbientLab, San Jose, CA) with dimensions of 36x27x10mm’.
Regarding the IMU’s location inside the shoe for the reproducibility of the experiments, the following sentence has been added in section 2.1: ‘In this way, the center of the IMU is positioned for a standard size 42 EU shoe, 8 mm below the top layer of the insole in contact with the foot (2 mm of insole thickness with 1 mm of 3D-printed PEBA-S and 5 mm as half of the IMU height) and 9 mm behind the heel center of impact, located at 12% of the sole length, here 32 mm (Figure 1).’
Figure 1 has been updated to show specific dimensions and the IMU location.
Comment 2: The data provided about the participants should include their gender (male, female) to ensure the reader can assess control over the data used in the study.
Response 2: Thanks for this remark. In the study, we did not adopt a gender-based approach but instead combined men and women. We have added in section 3.1 ‘Given the sample size, we decided to adopt a non-gender-specific approach in this study.’
Comment 3: How representative is the population used in the study? Why was this sample size chosen, and why not more? I believe 21 samples are limited. The selection of these samples must be properly justified.
Response 3: Regarding the representativeness of our sample in relation to a normal population, the following sentence has been added in section 3.1: ‘The average values of height, weight, and BMI can be considered typical averages for a normal population, with significant variability in the different characteristics.’
The choice of 21 participants comes from a comparison of our protocol with other existing similar step-counting protocols. Thus, we added the following sentence in section 2.2.1: ‘Similar studies have shown populations of varying sizes, ranging from 11 to 30 participants [19,35]. Therefore, the target in this study is set to recruit at least 20 participants’.
Comment 4: I recommend modifying Figure 1 to provide an enhanced technical diagram, clearly depicting the instrumentation to avoid confusion for the reader.
Response 4: We agree that the figure needs to be more technical. Therefore, Figure 1 has been modified by improving the technical information on the materials, thicknesses, and the positioning of the IMU inside the shoe.
Comment 5: I suggest adding a figure that includes an example of the recorded signal and an explanation of the data's meaning.
Response 5: Thank you for this suggestion. A figure (Figure 3: Raw accelerometer (left) and gyroscope (right) values associated with corresponding foot-status) has been added to section 2.2.4 to show an example of recorded signals and a visual representation of foot-status.
The following sentence has been added in section 2.2.4 to introduce the figure: ‘An example of 1 second of enriched data from raw acquisition values is shown in Figure 3.’
Comment 6: I recommend including a statement before the conclusion section, highlighting the novelty of the work compared to similar published studies.
Response 6: Thanks for this remark. To highlight the novelty and relevance of our approach, we have written in the discussion section: ‘Therefore, very good accuracy despite multi-activity and important variations in movement of subjects as compared to other studies.’
Comment 7: A paragraph should be added discussing potential future applications of the scientific proposal presented in the manuscript, such as monitoring different types of gaits in quadruped robots for control purposes [Biomimetics 2024, 9, 318].
Response 7: Thanks for this suggestion. To further expand the potential perspectives of this study, whether in human locomotion research or more generally in the monitoring of biomimetic robots, we added the following sentence at the end of the Discussion section: ‘Accurately detecting the steps in ecological multi-activity conditions creates opportunities for numerous future applications, including monitoring the efficiency of training programs, estimating injury risk, or can be extended to the monitoring of different types of gaits in biomimetic robots.’
Comment 8: It would be useful for readers if the authors include segments of code (or pseudocode) for the novel algorithms developed in this work.
Response 8: We agree with this comment. Therefore we have added the segment of code of the recurrent neural network used as Supplementary Material.
The following sentence has been added in section 3.2.2 to introduce the figure: ‘Those parameters are implemented in the RNN structure of the supplementary Python file S1.’
I would like to thank you for all your comments on this manuscript, and I hope I have addressed your concerns.
Reviewer 2 Report
Comments and Suggestions for Authors
This work using an existing running shoe boosted by inertial measurement unit (IMU) and used in multi-activity trials, forming a database, based on which a step counting algorithm powered by deep learning were tested. Results revealed the fine-tuned algorithm contributed to a MAPE of 0.75% ,making this approach suitable for embedded devices. The text is ok with figures and diagrams clearly presented. Still there are some issues for authors to consider for further improvement.
0. please re-write the abstract. Please do not put too many sentences of background, better to start with one brief intro, and cut into methods and results directly, followed by brief conclusion/significance.
1. It's expected that the location of the IMU may limit the transferability of the approach, as other studies have shown better performance with sensors placed on the wrist, waist, or thigh. Hence what is rational of using an IMU located under the heel for step counting accuracy? please clarify.
2. The work shows good results in multi-activity conditions. Reader may wonder whether or not can the algorithm be further refined for more diverse activities. Authors may discuss on this, cuz this could provide insights into its versatility and potential applications.
3.This work used 21 participants to form an experimental database. It's valuable for authors to discussing the diversity of the dataset, including participant and activity styles, which could help assess the generalizability of the findings and identify areas for future research.
Comments on the Quality of English Languagealready included in above.
Author Response
Thank you very much for taking the time to review this manuscript. Please find the detailed responses below and the corresponding revisions in the re-submitted files.
Comment 0: please re-write the abstract. Please do not put too many sentences of background, better to start with one brief intro, and cut into methods and results directly, followed by brief conclusion/significance.
Response 0: Thank you for pointing this out. The background section of the abstract has been rewritten to make it more concise. Therefore, we revised the background from 'A large range of locations of sensors and associated data analysis algorithms were previously experimented, with a best mean absolute percentage error (MAPE) close to 1% in simple mono-activity walking conditions.' to 'Various sensor placements and algorithms were previously experimented, with a best mean absolute percentage error (MAPE) close to 1% in simple mono-activity walking conditions.' This change allows the added elements of method details. The following sentence has been added to the abstract: 'A total of 21 participants with diverse profiles (gender, age, BMI, activity style) completed the trial.'.
Comment 1: It's expected that the location of the IMU may limit the transferability of the approach, as other studies have shown better performance with sensors placed on the wrist, waist, or thigh. Hence what is rational of using an IMU located under the heel for step counting accuracy? please clarify.
Response 1: We agree that the paragraph in the discussion was unclear and confusing, as it led the reader to believe that better results are achieved with sensors placed on the thigh instead of under the heel. The purpose of this paragraph is to highlight the influence of IMU location on step-counting performance while emphasizing the excellent performance obtained with a sensor placed under the heel and pointing out the limited transferability of our approach. This discussion paragraph has been entirely rewritten as follows: 'Another limitation of the study lies in the specific IMU location, underneath the heel. Other approaches found in the literature highlight the influence of IMU location on performance in mono-activity scenarios with devices easier to integrate on the wrist, waist, or thigh [42]. However, our study shows that placing the sensor underneath the heel achieves excellent performance in multi-activity scenarios. Our approach is therefore only transferable on the condition that the IMU is positioned in the same location.'
Comment 2: The work shows good results in multi-activity conditions. Reader may wonder whether or not can the algorithm be further refined for more diverse activities. Authors may discuss on this, cuz this could provide insights into its versatility and potential applications.
Response 2: Thanks for this comment pointing out the versatility of this algorithm. Therefore, we added the following sentence in the discussion section: 'Furthermore, since the algorithm is based on a deep learning approach, it is inherently capable of learning in new conditions, such as new activities, and can be further refined as soon as the model is enriched with data corresponding to these new conditions. '
Comment 3: This work used 21 participants to form an experimental database. It's valuable for authors to discussing the diversity of the dataset, including participant and activity styles, which could help assess the generalizability of the findings and identify areas for future research.
Response 3: We agree. The diversity of the dataset is now discussed with the addition of the following sentence in section 2.2.1 of the manuscript: ‘The participants are not representative of any particular category of the population, as they have different profiles in terms of age, height, weight, BMI, and activity levels.’
Information about physical activity was also added to both the methods and results sections. In the methods section 2.2.1, the following sentence was added: ‘The activity level is defined by their self-reporting of the weekly frequency of moderate-intensity activities (3.0-5.9 metabolic equivalents) and vigorous-intensity activities (6.0 and higher metabolic equivalents) [35].’
In the results section 3.1, the following sentence was added: ‘10 participants reported no moderate or vigorous physical activity, 8 participants reported an activity frequency of 1 to 2 times per week, 2 participants reported a frequency of 3 to 4 times per week and 1 participant reported an activity frequency of more than 5 times per week.’
I would like to thank you for all your comments on this manuscript, and I hope I have addressed your concerns.
Round 2
Reviewer 1 Report
Comments and Suggestions for Authors
The authors have addressed most of the suggestions satisfactorily. However, in comment 7, they forgot to include the reference in line 362 of the revised manuscript. The suggested reference to add is: [Acinonyx jubatus-Inspired Quadruped Robotics: Integrating Neural Oscillators for Enhanced Locomotion Control. Biomimetics 2024, 9, 318].
Author Response
Thanks for your feedback on the manuscript.
Comment 1: However, in comment 7, they forgot to include the reference in line 362 of the revised manuscript. The suggested reference to add is: [Acinonyx jubatus-Inspired Quadruped Robotics: Integrating Neural Oscillators for Enhanced Locomotion Control. Biomimetics 2024, 9, 318].
Response 1: As suggested, the reference is now included in the manuscrit in the sentence: 'Accurately detecting the steps in ecological multi-activity conditions creates opportunities for numerous future applications, including monitoring the efficiency of training programs, estimating injury risk, or can be extended to the monitoring of different types of gaits in biomimetic robots [43].'